# Ag$_2$Se as a tougher alternative to n-type Bi$_2$Te$_3$ thermoelectrics

Min Liu[1,2], Xinyue Zhang[1,2], Shuxian Zhang[1] & Yanzhong Pei [1]✉

For half a century, only Bi$_2$Te$_3$-based thermoelectrics have been commercialized for near room temperature applications including both power generation and refrigeration. Because of the strong layered structure, Bi$_2$Te$_3$ in particular for n-type conduction has to be texturized to utilize its high in-plane thermoelectric performance, leaving a substantial challenge in toughness. This work presents the fabrication and performance evaluation of thermoelectric modules based on n-type Ag$_2$Se paring with commercial p-Bi$_2$Te$_3$. Ag$_2$Se mechanically allows an order of magnitude larger fracture strain and thermoelectrically secures the module efficiency quite competitive to that of commercial one for both refrigeration and power generation within ± 50 K of room temperature, enabling a demonstration of a significantly tougher alternative to n-type Bi$_2$Te$_3$ for practical applications.

Thermoelectric technology enables a direct conversion between electricity and thermal energy for both power generation and cooling[1]. The performance is strongly dependent on the material's figure of merit $zT = S^2T/\rho\kappa$, in which $S$, $\rho$, $T$, and $\kappa$ are Seebeck coefficient, electrical resistivity, absolute temperature, and thermal conductivity respectively.

While progress has been notable in mid- and high-temperature materials, such as IV–VI compounds including PbTe[2], GeTe[3], and SnSe[4], skutterudites[5,6], Mg$_2$Sn[7], and half-Heuslers[8], advancements in near-room-temperature materials have been relatively slow. Bi$_2$Te$_3$ has stood out as the sole commercialized thermoelectric materials[9–12] for longer than half a century. Because of the strong layered structure, Bi$_2$Te$_3$ largely relies on texturization to utilize its high in-plane $zT$[13], leaving a substantial challenge in toughness.

Recently, several promising alternatives to commercial Bi$_2$Te$_3$ have been reported[10,11,14–16], such as Mg$_3$Sb$_2$[10,11,14], MgAgSb[14,15], and Ag$_2$Se[17,18] which exhibit compatible materials' $zT$ and offer greater sustainability. This motivated a few studies reporting impressive power generation efficiency and/or cooling performance using these materials, including n-Mg$_3$Sb$_2$/p-Bi$_2$Te$_3$[11], p-MgAgSb/n-Mg$_{3.2}$Bi$_{1.5}$Sb$_{0.5}$[14], n-Mg$_3$(BiSb)$_2$/p-MgAgSb[19], and n-Mg$_3$Sb$_2$/p-CdSb[20] combinations.

Ag$_2$Se was initially investigated as a thermoelectric material in the 1960s[21] and has since been improved to show a $zT$ above 0.7 near room temperature[22–25]. Ag$_2$Se undergoes a phase transition from the low-temperature orthorhombic phase to the high-temperature cubic phase at ~406 K[26]. The high $zT$ was usually realized in the orthorhombic phase

of Ag$_2$Se and was mainly attributed to its high carrier mobility and low lattice thermal conductivity[27]. However, phase transitions are typically undesirable as they may result in volume variations, which could lead to structural damage either within the material itself or at the interface between the material and electrodes during service. This somewhat limited the research on Ag$_2$Se to focus on exploring its material properties[23,25,28] and fabricating film devices specifically designed to operate at room temperature[17,18,29–31]. There are few reports on the power generation and cooling performance of Ag$_2$Se bulk modules. This motivates the current work to focus on exploring device properties of bulk Ag$_2$Se bellowing its phase transition temperature.

In addition to thermoelectric performance, mechanical properties are of equal importance to withstand loading. However, most inorganic thermoelectrics[32–35], including Bi$_2$Te$_3$[36] are intrinsically brittle of their strong bond ionicity and/or covalency, which is therefore challenging for durable serviceability. Fortunately, Ag$_2$Se[26,37] was found to show plasticity, which indicates a great advantage for tough thermoelectric applications near room temperature.

This work focuses on bulk Ag$_2$Se devices below its phase transition temperature. Ag$_2$Se bars with a high average $zT$ of 0.7 within 300–380 K were fabricated using a one-step hot-pressing process, to enable a contact resistivity as low as 12 μΩ cm$^2$ using Ni and Ag as electrodes. These electrode-bonded bars were then paired with commercial p-Bi$_2$Te$_3$ legs for fabricating modules (7 pairs, 12 × 12 mm), to enable a power generation efficiency $\eta_{max}$ of >1% at $\Delta T = 50$ K and a

[1]Interdisciplinary Materials Research Center, School of Materials Science and Engineering, Tongji Univ., 4800 Caoan Rd., Shanghai 201804, China. [2]These authors contributed equally: Min Liu, Xinyue Zhang. ✉e-mail: yanzhong@tongji.edu.cn

maximum cooling temperature $\Delta T_{max}$ of >50 K. In addition to offering the above device performance that is highly competitive to commercial Bi$_2$Te$_3$ modules, Ag$_2$Se has a distinct advantage in toughness indicated by its large fracture strain by order of magnitude. This offers the great potential of the Ag$_2$Se for efficient and durable thermoelectric applications near room temperature.

## Results and discussion

The details of material synthesis, module fabrication (dimensions in Table S1), characterizations, and property measurements (including the setup in Fig. S1) are given in the Methods. Three cylinders, with or without Ni/Ag electrodes, were hot-pressed in this work. As shown in Fig. 1, the hot-pressed cylinder without electrode is cut parallel (for XRD, $S$, $\rho$, $\kappa$ measurements, bending and compression tests) and perpendicular (for XRD, $S$, $\rho$, $\kappa$ measurements, and bending test) to the pressure direction. The cylinders with electrodes are sliced along the pressure direction for module fabrication.

X-ray diffraction (XRD) patterns for hot-pressed Ag$_2$Se pellets cut along directions perpendicular ($\perp$) and parallel (//) to that of pressure applied during hot pressing are shown in Fig. S2a. All the diffraction peaks can be well indexed to Ag$_2$Se of an orthorhombic structure, with no observable impurity peaks, thus affirming the purity of the samples. In addition, the similar relative intensities of diffraction peaks observed in Ag$_2$Se ($\perp$) and Ag$_2$Se (//) samples suggest the absence of obvious texture, indicating the transport-property isotropy in these polycrystalline materials. The scanning electron microscopy (SEM) images and the corresponding energy dispersive spectroscopy (EDS) mapping (Fig. S2b) further corroborate the high purity and homogeneity. The measured differential scanning calorimetry (DSC) curve (Fig. S2c) showing an endothermic peak at 406 K agrees well with the reported phase transition temperature of Ag$_2$Se[38].

Temperature-dependent thermoelectric transport properties for Ag$_2$Se ($\perp$) and Ag$_2$Se (//) samples were measured and shown in Fig. S3. Note that all measurements in this work were carried out under 380 K to avoid the phase transition at 406 K, which ensures the measurement repeatability. Both electronic and thermal transport properties of Ag$_2$Se ($\perp$) and Ag$_2$Se (//) are measured to be isotropic. The transport properties can be well described by a single parabolic band (SPB) model with an acoustic scattering (Fig. S3b, c). An average $zT$ of 0.7 within 300–380 K was achieved in Ag$_2$Se (//), which is comparable to available results of polycrystalline Ag$_2$Se[17,22,23,26,28,39,40] (Fig. S3f) and to that of commercial n-type Bi$_2$Te$_3$[41,42].

Vickers hardness, three-point bending, and compression tests were conducted on both commercial Bi$_2$Te$_3$ by an extrusion technique and as-fabricated Ag$_2$Se at room temperature to evaluate their mechanical properties. The Vickers hardness of Ag$_2$Se is measured to be ~35 kgf mm$^{-1}$ (Fig. S4a) along both perpendicular ($\perp$) and parallel (//) directions, which is consistent with the literature results[26,28]. The Vickers hardness of n- and p-type Bi$_2$Te$_3$ along the parallel (//) direction is measured to be 55 kgf mm$^{-1}$ and 48 kgf mm$^{-1}$, respectively, which is slightly higher than that measured along the perpendicular ($\perp$) direction. The difference can be understood by the orientation preference of the extruded materials as confirmed by XRD results and the calculated orientation factor $F$(110), which is 0.18 for n-type and 0.10 for p-type Bi$_2$Te$_3$, as shown in Fig. S5.

While the hardness of Ag$_2$Se may be inferior to that of Bi$_2$Te$_3$, its bending strength and compressive strength are significantly higher than those of Bi$_2$Te$_3$. This largely benefits machining and device operation. As shown in Fig. 2a, b, both n- and p-type commercial Bi$_2$Te$_3$ are unable to withstand bending strains above 0.5% or compressive strains above 2.5% and exhibit brittle fractures. Note that although zone-melted n-type Bi$_2$Te$_3$ has inferior mechanical properties compared to the p-type[43], the commercial Bi$_2$Te$_3$ used in this work was prepared by the hot extrusion technique. It has been proven that the mechanical properties of n-type Bi$_2$Te$_3$ are better than those of p-type[43]. In strong contrast, Ag$_2$Se allows additional large plastic deformation (Fig. 2c) to enable an ultimate bending strain of 4% at 128 MPa. Similarly, during the compression test (Fig. 2d), Ag$_2$Se exhibits an elastic strain of 2% before yielding at ~50 MPa, followed by plastic deformation to enable a final strain as large as 40% at 273 MPa. The significant difference in mechanical properties can also be clearly reflected by the microstructures of the fracture surface (Fig. S6). Intergranular crack paths are observed at the fracture surface of Bi$_2$Te$_3$ during both bending and compression tests, indicating brittle fracture. In the case of Ag$_2$Se, although macroscopic cracks are observed at the surface, the bulk sample is not completely fractured, demonstrating excellent toughness.

To optimize the module performance, minimizing contact resistance is essential, which ensures a low device internal resistance ($R_{in}$), thereby maximizing both the output power ($P_{max}$) and conversion efficiency ($\eta$) of the device. In this work, both Ni and Ag were found to facilitate low contact resistance when directly bonded to Ag$_2$Se by hot-pressing. The electrical contact resistance ($R_c$) at Ag$_2$Se/Ni (module 1) and Ag$_2$Se/Ag (module 2) joints are estimated to be 0.3 m$\Omega$ (Figs. S7a and 7b). This corresponds to an interfacial contact resistivity ($\rho_c$) as low as 12 μ$\Omega$ cm$^2$, leading the total contact resistance to be ≤10% of the $R_{in}$ of the leg. A robust bonding without any cracks is confirmed by SEM observations taken after the hot pressing and welding processes (Fig. S8). The total resistances of these modules are shown in Fig. S9.

With a fixed cold side temperature of ~285 K for power generation, the open-circuit voltage $V_{OC}$, maximum output power $P_{max}$, heat flow $Q$, and maximum conversion efficiency $\eta_{max}$ versus different

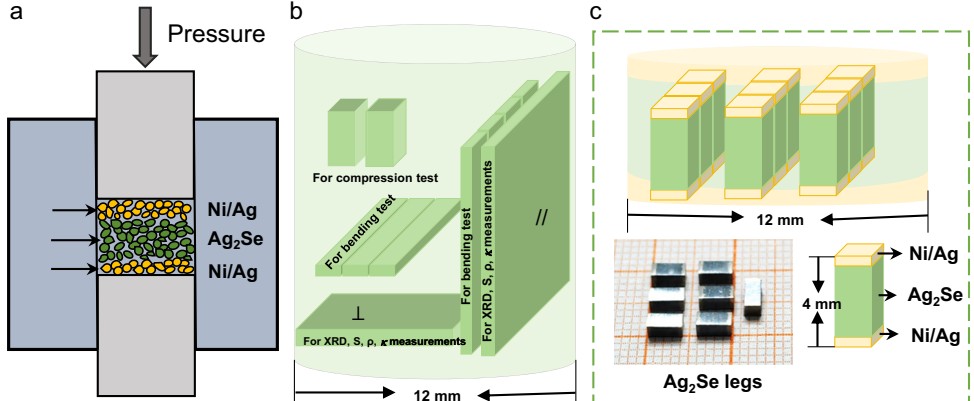

**Fig. 1 | Schematic of the fabrication and slicing. a** Schematic of the one-step hot-pressing, schematic slicing diagram of **b** the cylinder for property measurements and **c** the cylinder with electrodes for module assembly. Photograph of Ag$_2$Se legs and the contact structure are also shown (**c**).

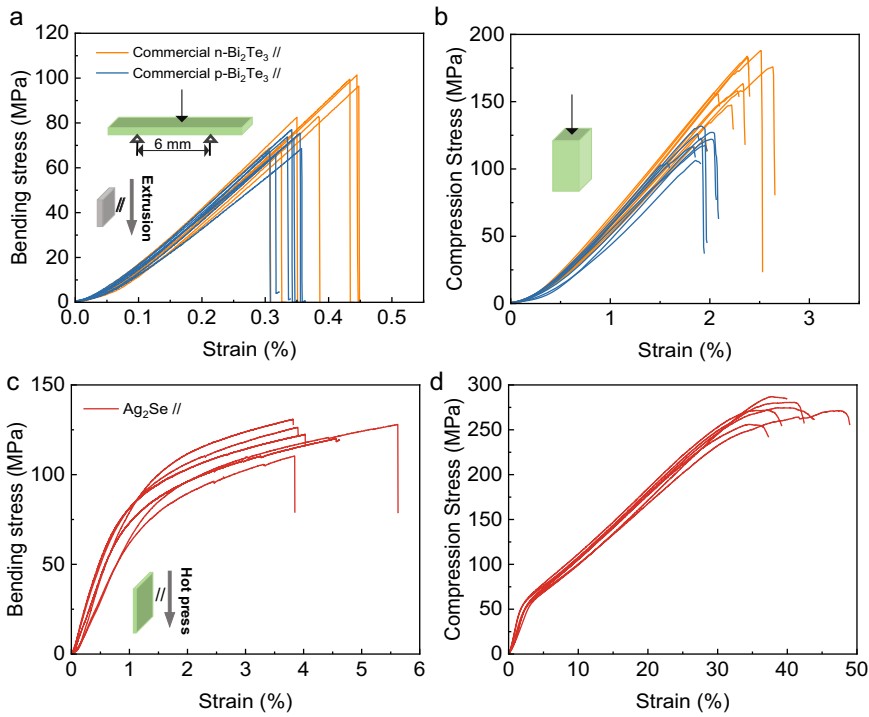

**Fig. 2 | Mechanical properties. a**, **c** Stress–strain curves for bending and **b**, **d** for compression tests for commercial $Bi_2Te_3$ by extrusion and as-fabricated $Ag_2Se$ by hot-pressing at room temperature.

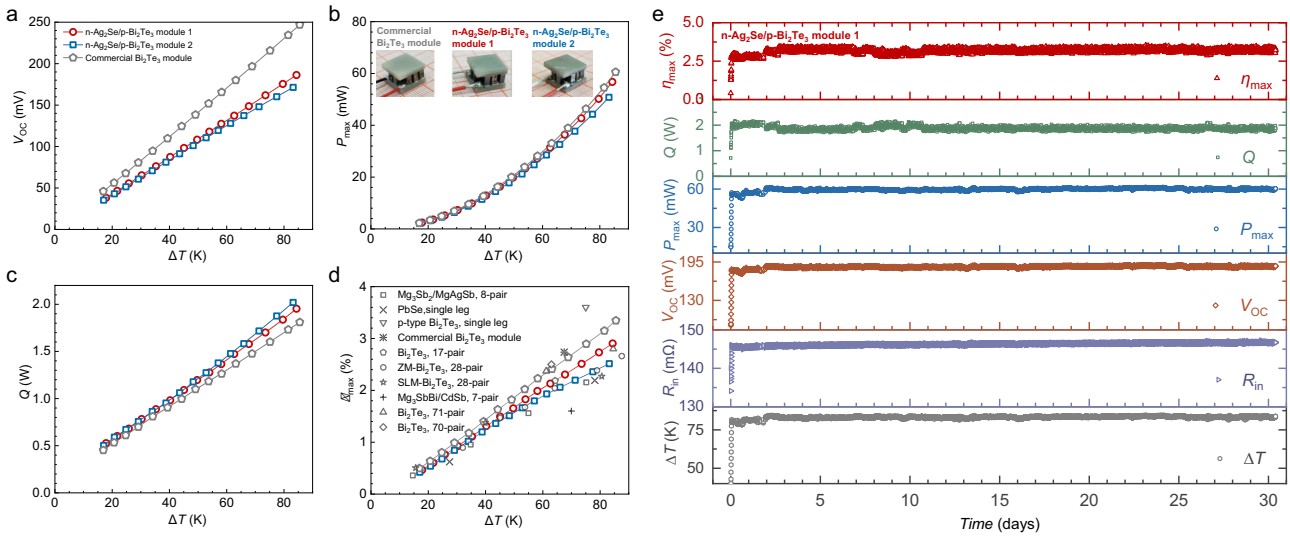

**Fig. 3 | Power generation performance. a** Open-circuit voltage ($V_{OC}$), **b** maximum output power ($P_{max}$), **c** heat flow ($Q$), and **d** conversion efficiency ($\eta_{max}$) as a function of different temperature gradients ($\Delta T$) for n-$Ag_2Se$/p-$Bi_2Te_3$ modules and commercial $Bi_2Te_3$ one. Literature results are included for comparison[9,10,15,16,20,42,47–49], **e** $\eta_{max}$, $Q$, $P_{max}$, $V_{OC}$, internal resistance $R_{in}$ and $\Delta T$ of module 1 during continuous measurements for 30 days at $\Delta T$ of ~85 K.

temperature gradients ($\Delta T$) are shown in Fig. 3 for both modules and commercial $Bi_2Te_3$ module. The measured voltage $V$, $P$, and $Q$ as a function of current $I$ for these modules at different $\Delta T$ are presented in Fig. S10. It can be seen that both modules in this work enable quite competitive performance with that of the commercial one, particularly within a $\Delta T$ of 50 K. Long-term efficiency measurements at a $\Delta T$ of ~85 K are performed on the modules to check the thermal stability. Although the linear coefficient of thermal expansion (CTE) of $Ag_2Se$ differs from p-$Bi_2Te_3$ at 300–373 K (Fig. S11), no obvious degradation in $\eta_{max}$, $P_{max}$, $V_{OC}$, and $R_{in}$ is observed for module 1 (Ni electrode) after continuous measurement for 30 days (Fig. 3e). However, the $R_{in}$ of module 2 (Ag electrode) increases obviously as the measurement time

progresses (Fig. S12), leading to a notable degradation in both output power and efficiency. This indicates that using Ni as electrodes for $Ag_2Se$ enables superior long-term stability of the module compared to using Ag as electrodes.

The cooling capability of both modules at a fixed hot-side temperature of ~300 K is shown in Fig. 4. The measured cold-side temperatures ($T_c$) under different input current $I$ are provided in Fig. S13. The maximum cooling temperature difference $\Delta T_{max}$ reaches ~56 K, which is quite comparable to that of a commercial one. Similar comparability (Fig. 4a–c) can also be seen from the measurements on current dependent maximum cooling power and coefficient of performance ($COP$), as well as on $\Delta T$ dependent maximum coefficient of

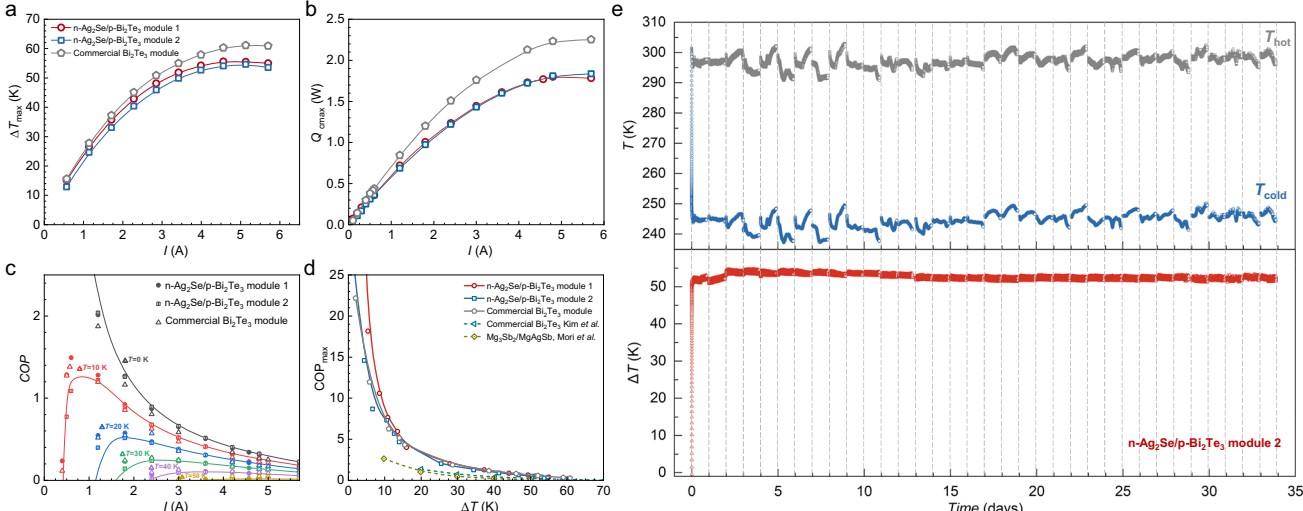

**Fig. 4 | Cooling performance. a** measured maximum cooling temperature difference ($\Delta T_{max}$), **b** current dependent maximum cooling power ($Q_{cmax}$), and **c** coefficient of performance (COP), as well as **d** maximum COP as a function of different $\Delta T$ for the modules[50], **e** cold-side temperature ($T_c$), hot-side temperature ($T_h$) and corresponding cooling temperature difference ($\Delta T$) of n-Ag$_2$Se/p-Bi$_2$Te$_3$ module 2 at a given current $I$ of 4.3 A during duration measurements for 34 days, showing good stability. During the cooling duration tests, we run the test once per day for a continuous period of 22 h.

performance (COP$_{max}$), indicating equivalent energy consumption while pumping similar amount of heats. Although module 2 (Ag electrode) showed unsatisfactory stability during long-term power generation measurements, its cooling performance was found to be quite stable after 34 days of measurements (Fig. 4e). Note that module 1 performs better than module 2 in both power generation and cooling applications. Since the contact resistivity of Ag$_2$Se/Ni and Ag$_2$Se/Ag are very close (Fig. S7), the reason for the better performance of module 1 is presumed to be that the Ag$_2$Se in module 1 has a slightly lower carrier concentration that is closer to the optimal value. This is evidenced by the higher voltage output and internal resistance of module 1 (Fig. S10). In addition, the device performance still has room for further improvement through geometric optimization according to the numerical simulation (Fig. S14).

In summary, bulk Ag$_2$Se is ensured to show quite comparable near-room-temperature thermoelectric properties to commercial n-Bi$_2$Te$_3$, at both material and device levels for both power generation and cooling applications. Long-term measurements for over one month demonstrate that Ni is a better choice than Ag as electrodes for Ag$_2$Se, ensuring good thermal stability. Most importantly, mechanical evaluations demonstrate the much more superior toughness in Ag$_2$Se, highlighting the great capability to address the historical challenge for durable and large-scale thermoelectric applications near room temperature.

## Methods
### Synthesis
Polycrystalline Ag$_2$Se was synthesized using high purity (>99.99%) of Ag and Se granules, weighted according to the stoichiometric ratio of Ag:Se = 2:1, loaded into the quartz tubes and sealed under vacuum. The raw materials were heated to 1273 K in 7 hours and kept at this temperature for 8 hours, then cooled down to 773 K in 5 hours and kept 773 K for 48 h followed by furnace cooling to room temperature. The obtained ingots were grinded and then densified by hot pressing. Dense Ag$_2$Se pellets, Ni/Ag$_2$Se/Ni, and Ag/Ag$_2$Se/Ag cylinders with ~12 mm in diameter were obtained by hot pressing at 573 K for 20 minutes under a uniaxial pressure of ~60 MPa. As shown in Fig. 1, the Ni/Ag$_2$Se/Ni and Ag/Ag$_2$Se/Ag legs were sliced from the obtained Ni/Ag$_2$Se/Ni and Ag/Ag$_2$Se/Ag cylinders with a dimension of $2 \times 2 \times 4$ mm$^3$, the same size of legs in commercial Bi$_2$Te$_3$ modules. We

used low-temperature solder (In$_{52}$Sn$_{48}$, with a melting point of 391 K) to avoid phase transition. The soldering process was conducted at 391 K, which does not reach the phase transition temperature of Ag$_2$Se (~406 K).

### Characterization and transport-property measurements
XRD and transport properties were measured on samples sliced along the directions parallel and perpendicular to that of pressure applied during hot pressing, as shown in Fig. 1b. The orientation factor $F$ of (110) was calculated from the ratios of the integral intensities of the (110) planes to the intensities of the (hkl) planes for preferentially and for randomly orientated samples according to the Lotgering method[44]. The phase transition of Ag$_2$Se was confirmed by a differential scanning calorimetry (DSC) measurement system (Netzsch DSC 3500 Sirius). The phase composition and microstructures of the materials were characterized by X-ray diffraction (XRD, DX-2700) and a scanning electron microscope (SEM, Phenom Pro, and Zeiss Sigma 300VP) equipped with an energy dispersive spectrometer (EDS).

Resistivity, Hall mobility, carrier concentration, and Seebeck coefficient were simultaneously measured at various temperatures under helium. The details of the measurements can be found in our previous work[45]. The thermal conductivity was calculated by $\kappa = dC_P\lambda$, where $d$ is the density measured by Archimedes drainage method, $C_p$ is the specific heat, $\lambda$ is the thermal diffusivity measured using laser flash technique (Netsch LFA 467). Sound velocities were measured at room temperature using a pulse receiver (Olympus-NDT) equipped with an oscilloscope (Keysight). Water and Shear gel (Olympus) was used as couplant during the measurements of longitudinal and transverse sound velocity ($v_L$ and $v_S$), respectively.

### Mechanical property
Vickers hardness test was carried out using Vickers hardness tester (DHV-1000) under the load of 2.94 N holding for 10 s. Mechanical tests, including three-point bending and compression tests, were performed on hot-pressed Ag$_2$Se and commercially extruded Bi$_2$Te$_3$ (Xiamen X-Meritan Technology Co., Ltd.) using a micro-computer controlled electronic universal test machine [Lishi (Shanghai) Instruments Co., Ltd., P. R. China] with loading rates of 0.03 mm/min and 0.5 mm/min, respectively. During three-point bending tests, the span of the fixture was kept at 6 mm, while the sheet was 2.5 mm in width and 1 mm in

thickness. Bulks with the size of $2 \times 2 \times 4$ mm$^3$ were used for compression tests, the same direction and dimensions as device legs (Fig. 1). The yielding strength ($\sigma_{0.2}$) in bending and compression were the bending and compressive 0.2% offset stress from the bending and compressive stress-strain curve, respectively. After bending and compression tests, the surfaces and fracture surfaces were characterized by SEM with a secondary electron (SE) detector (Zeiss Sigma 300VP).

### Conversion efficiency and cooling performance

The commercial Bi$_2$Te$_3$ module used in this work is from Xiamen X-Meritan Technology Co., Ltd. (7 pairs, $12 \times 12$ mm$^2$), the detailed parameters of which are listed in Table. S1. The n-Ag$_2$Se/p-Bi$_2$Te$_3$ modules in this work were fabricated by replacing the n-type Bi$_2$Te$_3$ legs of commercial Bi$_2$Te$_3$ modules with n-type Ag$_2$Se legs.

The contact resistance ($R_c$) was measured by a four-probe technique at a constant electric current of -200 mA. The interfacial contact resistivity ($\rho_c$) was estimated by $\rho_c = R_c \times A$, where $A$ is the cross-section area of the leg. The total resistance of the module was obtained from the slope of the AC voltage vs. current between the two copper wires of the modules within 0–0.2 A.

A homemade power-generation measurement system was used to measure conversion efficiency (Fig. S1a). The cold-side temperature ($T_c$) was maintained by a water-cooling system. K-type thermocouples were used to measure the hot side ($T_h$) and cold side ($T_c$) temperatures of modules and the temperature difference of the heat flow meter ($\Delta T_{Cu}$). The output power $P$, heat flow $Q$, and conversion efficiency $\eta$ of the modules under different temperature gradients were measured in a vacuum. $P$ is determined by $P = IV$, where $I$ is the current, and $V$ is the output voltage. $Q$ is obtained by $Q = (\kappa_{Cu} A_{Cu} \Delta T_{Cu})/L_{Cu}$, where $\kappa_{Cu}$, $A_{Cu}$, $\Delta T_{Cu}$, and $L_{Cu}$ are thermal conductivity, cross-section area, temperature difference, and distance between thermocouples of the heat flow meter. The average $\kappa_{Cu}$ used for determining $Q$ is ~386 W m$^{-1}$ K$^{-1}$[46]. $A_{Cu}$ and $L_{Cu}$ are $12 \times 12$ mm$^2$ and 25 mm, respectively. Therefore, $\eta$ is determined by $\eta = P/(P + Q)$. The maximum $\eta$ at a given $T_h$ can be obtained by varying the load resistance in the circuit. We measured each parameter (including temperature, voltage and current) for 30 times to minimize the system error.

For cooling performance measurement (Fig. S1b), the thermoelectric module was attached to Cooper Block with a built-in cooling water circulation system using thermal silicone grease. Two K-type thermocouples were used to record temperatures ($T_h$ for hot-side, and $T_c$ for cold-side temperature). A ceramic heater with a similar size ($12 \times 12$ mm$^2$) was attached to the top of the module using the same silicone grease to measure the cooling power $Q_C$ at various temperature reductions $\Delta T = T_h - T_c$. The minimum temperature of the cold side ($T_c^{min}$) can be obtained by varying the input current $I$ in the circuit, and the maximum cooling temperature difference ($\Delta T_{max} = T_h - T_c$) was estimated at equilibrium when the heater did not work. All measurements were carried out under the vacuum of <1 Pa with $T_h$ of ~297-300 K. The coefficient of performance ($COP$) is determined by $COP = Q_C/P_{in}$, where $Q_C$ is cooling power, and $P_{in}$ is total input power. $Q_{Cmax}$ is the maximum cooling power under a given current $I_q$ when $\Delta T_{max} = 0$ K.

### Data availability

The authors declare that all data supporting the findings of this study are available within the article and its Supplementary Information files or from the corresponding author upon request.

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

## Acknowledgements

Y.Z.P. acknowledges the National Natural Science Foundation of China (grant numbers T2125008, 92163203) and the Innovation Program of Shanghai Municipal Education Commission (2021-01-07-00-07-E00096). X.Y.Z. acknowledges the National Natural Science Foundation of China (grant number 52102292). M.L., Y.Z.P., X.Y.Z., and S.X.Z. acknowledge the Fundamental Research Funds for the Central Universities. The authors thank Prof. Bo Chen from Tongji University for the support on SEM characterization. The authors thank Prof. Jia Huang from Tongji University for the support on compression and bending tests.

## Author contributions

Y.Z.P. conceived the idea and supervised the project. M.L. prepared samples. M.L. and X.Y.Z. analyzed data and wrote the paper. S.X.Z. helped with the power generation analysis. All the authors discussed the content of the paper.

## Competing interests

The authors declare no competing interests.
