## [Peer Review File · Nature Communications]

Ag₂Se as a tougher alternative to n-type Bi₂Te₃ thermoelectricsREVIEWER COMMENTS

Reviewer #1 (Remarks to the Author):

The authors fabricated the bulk Ag₂Se device for the first time and then investigated the cooling and power generation efficiencies. The results show Ag₂Se has quite comparable near-room-temperature thermoelectric properties to commercial n-Bi₂Te₃ at both material and device levels. This study is important and very interesting, and thus deserve to be published in Nature Communications. Nevertheless, there are also some concerns regarding some specific issues.

- 1) The expression of zT is wrong in the introduction part.
- 2) The reported alternative to commercial Bi₂Te₃ by Zhao should be cited in the introduction. [Science 383.6688 (2024): 1204-1209].
- 3) The detailed constituent for p-Bi₂Te₃ legs in your device as well as its electrodes should be provided for readers to repeat the results.
- 4) Although there are many papers on high performance Ag₂Se thermoelectrics (having ZT above 1 near room temperature), almost no report on bulk Ag₂Se thermoelectric device including the single leg [Energy Environ. Sci., 2023, 16, 1870–1906]. What's the key issue that hinder the development of bulk Ag₂Se device? In order to highlight the importance of this work, this problem needs to be discussed in the introduction as well as your solution.
- 5) As we know, the performance of Ag₂Se is sensitive to defects like Ag interstitial and also the ratio of Ag/Se [Energy Environ. Sci., 2023, 16, 1870–1906], and the mass transport/the reaction of defects will happen between beta-Ag₂Se and Ag electrode at high temperature sintering (673 K, above phase transition temperature) processing. One can also observe the content of Se is obviously higher in the side of silver electrode when compared with nickel electrode in Fig S7c-d, which possibly support the above-mentioned reaction. How to solve the problem of the interfacial reaction in high temperature welding for this superionic conductor Ag₂Se at high temperature, which is important for the application of Ag₂Se and needed to be discussed.
- 6) (i)The problem of the detrimental phase transition during cooling process after high temperature welding, which may cause thermomechanical stresses/crack and needed to be discussed; (ii) As thermal stress is unavoidable during the cooling process after high temperature welding, a similar coefficients of thermal expansion (CTE) values are needed to prevent the failure of Ag₂Se modules, please provide the temperature-dependent CTE of Ag₂Se, Bi₂Te₃ and also the electrodes (like Ag, Ni).
- 7) Module 1 owns better performance in both power generation and cooling applications (Figure 3 and 4) than that of module 2, why? The reason for the decreased performance in module 1 with silver electrode need to be explained, the interfacial reaction? Or others.

Reviewer #2 (Remarks to the Author):

The manuscript titled "Ag₂Se as a tougher alternative to n-type Bi₂Te₃ thermoelectrics" describes the following important issues. Ag₂Se is one of the best materials for near room temperature application due to the fact that its figure-of-merit is very high in temperature range of 300-380 K. I fully agree with the authors justification that it has better mechanical strength than Bi₂Te₃, which is very important from device fabrication point of view. My only worry is the long term stability of the n-Ag₂Se/p-Bi₂Te₃ based thermoelectric power generation module. In Ag₂Se it is well known that even at room temperature Ag⁺ ions can migrate and form clusters which can change the properties. The authors should test that long

term stability of the thermometric generator by measuring its output power continuously for one month and add that data in the manuscript to show the stability of Ag₂Se as well as Ag/Ag₂Se & Ni/Ag₂Se electrical contacts at high temperature. This additional experiment will really prove the real potential of Ag₂Se over Bi₂Te₃.

Reviewer #3 (Remarks to the Author):

This work presents the fabrication and performance evaluation of thermoelectric modules based on n-type Ag₂Se. While recent thermoelectric research on Ag₂Se has primarily focused on its film performance and devices, I find this study's emphasis on Ag₂Se bulk modules to be particularly intriguing. This work systematically investigated the toughness of bulk Ag₂Se compared to Bi₂Te₃, which is interesting and informative for future researches. The results demonstrate that Ag₂Se, with its superior bending and compressive strength, indeed represents a tougher alternative to n-Bi₂Te₃. The contact resistivity here is low. I believe this may mark the first demonstration as I didn't find any literature reporting contact resistivity in bulk Ag₂Se device. The n-Ag₂Se/p-Bi₂Te₃ module seems to be the first demonstration as well, and the performance is good. With these important results, I recommend that this work be published in Nature Communications after addressing the technical concerns

1. Paragraph 3 in part "Results and discussion": ".....The transport properties can be well described by a single parabolic band (SPB) model with an acoustic scattering (Figs. S3b and S3c).....". It is better to include the logarithmic relationship between temperature and mobility in Fig. S3 to illustrate acoustic scattering in Ag₂Se.
2. We generally consider n-type Bi₂Te₃ to be more brittle compared to p-type Bi₂Te₃. Why is it that here, the n-type Bi₂Te₃ exhibits higher bending and compressive strength?
3. Geometry design is also important for maximizing the performance of thermoelectric module. Why did the authors select a leg size of 2x2x4 mm³? Please provide further explanation.
4. Fig. S8 is not mentioned in the manuscript or the supplementary.
5. The internal resistances of the Ag₂Se-based module and the commercial Bi₂Te₃ module differ. However, the optimized currents for these two modules appear to be the same, as shown in Figure 4a and Figure S8. Any explanation on this issue?

Responses to the reviewers' comments

Reviewer #1 (Remarks to the Author):

The authors fabricated the bulk Ag₂Se device for the first time and then investigated the cooling and power generation efficiencies. The results show Ag₂Se has quite comparable near-room-temperature thermoelectric properties to commercial n-Bi₂Te₃ at both material and device levels. This study is important and very interesting, and thus deserve to be published in Nature Communications. Nevertheless, there are also some concerns regarding some specific issues.

Response: Thank you for the appreciation and constructive comments on this work. Below we address the reviewer's comments point by point. In addition to the revision requested by all reviewers, we retested the power generation performance of all modules since we found that the internal resistance of all modules had been overestimated in the previous power generation tests. During revision, we optimized the contact resistance of all connection points in the external circuit of the testing system, thereby improving test quality.

1) The expression of zT is wrong in the introduction part.

Response and revision: The typo has been corrected to $zT = S^2T/\rho\kappa$.

2) The reported alternative to commercial Bi₂Te₃ by Zhao should be cited in the introduction. [Science 383.6688 (2024): 1204-1209].

Response and revision: Thanks for the suggestion. The literature has been cited as ref.16.

3) The detailed constituent for p-Bi₂Te₃ legs in your device as well as its electrodes should be provided for readers to repeat the results.

Response and revision: Thanks for the suggestion. The p-type Bi₂Te₃ legs used in this work are the same as those in the commercial modules. We desoldered the commercial modules and used their p-Bi₂Te₃ legs to pair with our fabricated n-Ag₂Se legs to form our modules. Since the detailed constituent of these p-Bi₂Te₃ legs are proprietary, we provided information about the source and the model number of the involved commercial module (Table S1 in the supplementary information).

Table S1 Details for n-Ag₂Se/p-Bi₂Te₃ modules in this work and the commercial Bi₂Te₃ one.

Module	P-type	N-type	Pairs	Module size (mm ³)	Size of each leg (mm ³)
Commercial Bi ₂ Te ₃ module	P-Bi ₂ Te ₃ alloys, Model No. TEG1-712-0.14, Xiamen X-Meritan Technology Co., LTD.	N-Bi ₂ Te ₃ alloys, Model No. TEG1-712-0.14, Xiamen X-Meritan Technology Co., LTD.	7	12×12×8.2	2×2×4
n-Ag ₂ Se/p-Bi ₂ Te ₃ module 1	P-Bi ₂ Te ₃ alloys, Model No. TEG1-712-0.14, Xiamen X-Meritan Technology Co., LTD.	This work, Ag ₂ Se with Ni electrode	7	12×12×8.2	2×2×4
n-Ag ₂ Se/p-Bi ₂ Te ₃ module 2	P-Bi ₂ Te ₃ alloys, Model No. TEG1-712-0.14, Xiamen X-Meritan Technology Co., LTD.	This work, Ag ₂ Se with Ag electrode	7	12×12×8.2	2×2×4

4) Although there are many papers on high performance Ag₂Se thermoelectrics (having ZT above 1 near room

temperature), almost no report on bulk Ag₂Se thermoelectric device including the single leg [Energy Environ. Sci., 2023, 16, 1870–1906]. What's the key issue that hinder the development of bulk Ag₂Se device? In order to highlight the importance of this work, this problem needs to be discussed in the introduction as well as your solution.

Response and revision: Thanks for the suggestion. Ag₂Se undergoes a phase transition from low-temperature orthorhombic phase to the high-temperature cubic phase at ~406 K²⁶. The high zT was usually realized in the orthorhombic phase of Ag₂Se and was mainly attributed to its high carrier mobility and low lattice thermal conductivity²⁷. However, phase transitions are typically undesirable as they may result in volume variations, which could lead to structural damage either within the material itself or at the interface between the material and electrodes during service. This somewhat limited the researches on Ag₂Se to focus on exploring its material properties^{23, 25, 28} and fabricating film devices specifically designed to operate at room temperature^{17-18, 29-31}. There are few reports on power generation and cooling performance of Ag₂Se bulk modules. This motivates the current work to focus on exploring device properties of bulk Ag₂Se bellowing its phase transition temperature.

We revised to include above discussion in the introduction (line 28, page 1) with citation of the mentioned literature [Energy Environ. Sci., 2023, 16, 1870–1906, ref.31].

5) As we know, the performance of Ag₂Se is sensitive to defects like Ag interstitial and also the ratio of Ag/Se [Energy Environ. Sci., 2023, 16, 1870–1906], and the mass transport/the reaction of defects will happen between beta-Ag₂Se and Ag electrode at high temperature sintering (673 K, above phase transition temperature) processing. One can also observe the content of Se is obviously higher in the side of silver electrode when compared with nickel electrode in Fig S7c-d, which possibly support the above-mentioned reaction. How to solve the problem of the interfacial reaction in high temperature welding for this superionic conductor Ag₂Se at high temperature, which is important for the application of Ag₂Se and needed to be discussed.

Response and revision: Thank you for the comment. The previous SEM and EDS results of the Ag₂Se/Ag and Ag₂Se/Ni joints were obtained using traditional SEM equipment (Phenom Pro). Since traditional SEM is not as accurate as field emission SEM (FE-SEM) for elemental quantitative analysis, we conducted elemental analysis using FE-SEM equipment (Zeiss Sigma 300VP) to further confirm the precise content of the elements in the joints. It can be seen in updated Fig. S7c-f, Se was not detected in either the Ni electrode or the Ag electrode after hot pressing. However, the long-term measurements on two modules (Fig. 3e and Fig. S12) indeed indicate that Ag as an electrode for Ag₂Se is not a good choice for power generation concerning long-term stability. Using Ni as electrodes for Ag₂Se enables more stable power generation, as demonstrated by continuous measurements over 30 days (Fig. 3e). We updated Fig. S7 in the supplementary information and added a brief discussion about the long-term stability of modules using Ni or Ag as electrodes (line 93, page 2).

Fig. S7 Contact structure and resistance. Scanning resistance (R) across Ni/Ag₂Se/Ni (a) and Ag/Ag₂Se/Ag (b) junctions. SEM images, EDS mapping, EDS line scanning and elemental analysis for the Ag₂Se/Ni (c, e) and the Ag₂Se/Ag (d, f) joints.

6) (i) The problem of the detrimental phase transition during cooling process after high temperature welding, which may cause thermomechanical stresses/crack and needed to be discussed; (ii) As thermal stress is unavoidable during the cooling process after high temperature welding, a similar coefficients of thermal expansion (CTE) values are needed to prevent the failure of Ag₂Se modules, please provide the temperature-dependent CTE of Ag₂Se, Bi₂Te₃ and also the electrodes (like Ag, Ni).

Response and revision: In this work, we used low-temperature solder (In₅₂Sn₄₈, with a melting point of 391 K) to avoid phase transition. The soldering process was conducted at 391 K, which does not reach the phase transition temperature of Ag₂Se (~406 K). During this process, Ag₂Se does not undergo phase transition. In the preliminary step before soldering, the hot pressing of Ag₂Se legs was indeed conducted at high temperature (573 K). We performed SEM and EDS analysis on the cross-sections of the Ag₂Se legs that had undergone both hot pressing and soldering processes. A robust bonding without any cracks is confirmed by SEM observations taken after the hot pressing and welding processes (Fig. S8). We revised to include relevant clarification in the main text (line 74, page 2) and supplementary information (Synthesis section).

We provided CTE for n-Ag₂Se, Ni, Ag and p-Bi₂Te₃ involved in our modules, and added a brief discussion in the main text (line 93, page 2): Although the linear coefficient of thermal expansion (CTE) of Ag₂Se differs from p-Bi₂Te₃ at 300-373 K (Fig. S11), no obvious degradation in η_{max} , P_{max} , V_{OC} and R_{in} is observed for module 1 (Ni electrode) after continuous measurement for 30 days (Fig. 3e). However, the R_{in} of module 2 (Ag electrode) increases obviously as the measurement time progresses (Fig. S12), leading to a notable degradation in both output power and efficiency. This indicates that using Ni as electrodes for Ag₂Se enables superior long-term stability of the module compared to using Ag as electrodes.

Fig. S8 Contact structure after welding. SEM images, EDS mapping and EDS elemental analysis for the $\text{Ag}_2\text{Se}/\text{Ni}/\text{solder}$ (a) and the $\text{Ag}_2\text{Se}/\text{Ag}/\text{solder}$ (b) joints.

Fig. S11 Thermal expansion measurements. Temperature dependent relative length variation (dL/L_0) for $\text{n-Ag}_2\text{Se}$, Ni, Ag and $\text{p-Bi}_2\text{Te}_3$ involved in our modules. The values on the dL/L_0 curves represent the linear coefficients of thermal expansion (CTE) in the specific temperature range. The inset shows the measured CTE at 300-373 K with comparison to literature results [5], [18-19].

7) Module 1 owns better performance in both power generation and cooling applications (Figure 3 and 4) than that of module 2, why? The reason for the decreased performance in module 1 with silver electrode need to be explained, the interfacial reaction? Or others.

Response and revision: Since the contact resistivity of $\text{Ag}_2\text{Se}/\text{Ni}$ and $\text{Ag}_2\text{Se}/\text{Ag}$ are very close (Fig. S7), the reason for the better performance of module 1 is presumed to be that the Ag_2Se in module 1 has a lower carrier concentration, which is closer to the optimal value. This is evidenced by the higher voltage output and internal resistance of module 1 (Fig. S10). We revised the manuscript to include above discussion (line 21, page 3).

Reviewer #2 (Remarks to the Author):

The manuscript titled "Ag₂Se as a tougher alternative to n-type Bi₂Te₃ thermoelectrics" describes the following important issues. Ag₂Se is one of the best materials for near room temperature application due to the fact that its figure-of-merit is very high in temperature range of 300-380 K. I fully agree with the authors justification that it has better mechanical strength than Bi₂Te₃, which is very important from device fabrication point of view. My only worry is the long term stability of the n-Ag₂Se/p-Bi₂Te₃ based thermoelectric power generation module. In Ag₂Se it is well known that even at room temperature Ag⁺ ions can migrate and form clusters which can change the properties. The authors should test that long term stability of the thermometric generator by measuring its output power continuously for one month and add that data in the manuscript to show the stability of Ag₂Se as well as Ag/Ag₂Se & Ni/Ag₂Se electrical contacts at high temperature. This additional experiment will really prove the real potential of Ag₂Se over Bi₂Te₃.

Response and revision: We sincerely thank the reviewer for the positive comments. In order to address the reviewer's concern regarding the long-term stability, we have measured both the power generation and cooling performance of the modules for over one month. During this process, we optimized the contact resistance of all connection points in the external circuit of the testing system, thereby improving test quality. We found that the internal resistance of all modules had been overestimated in the previous power generation tests. Therefore, we retested the power generation performance of all modules. The long-term measurement results are shown in Figs. 3e, 4e, and S12. No obvious degradation in η_{\max} , P_{\max} , V_{OC} and R_{in} is observed for module 1 (Ni electrode) after continuous measurement for 30 days (Fig. 3e). However, the R_{in} of module 2 (Ag electrode) increases obviously as the measurement time progresses (Fig. S12), leading to a notable degradation in both output power and efficiency. This indicates that using Ni as electrodes for Ag₂Se enables superior long-term stability of the module compared to using Ag as electrodes. Although module 2 showed unsatisfactory stability during long-term power generation measurements, its cooling performance was found to be quite stable after 34 days of measurements (Fig. 4e). We revised to include Fig. 3e, Fig. 4e and above discussion in main text (line 96, page 2 and line 12, page 3), and include Fig. S12 in supplementary information.

Fig. 3 Power generation performance. (a) Open-circuit voltage (V_{OC}), (b) maximum output power (P_{\max}), (c) heat flow (Q) and (d) maximum conversion efficiency (η_{\max}) as a function of different temperature gradients (ΔT) for n-Ag₂Se/p-Bi₂Te₃ modules and commercial Bi₂Te₃ one. Literature results are included in (d) for comparison^{9-10, 15-16, 20, 42, 44-46}. (e) η_{\max} , Q , P_{\max} , V_{OC} , internal resistance R_{in} and ΔT of module 1 during continuous measurements for 30 days at ΔT of ~ 85 K.

Fig. S12 Power generation duration of module 2. Maximum conversion efficiency η_{\max} , heat flow Q , maximum output power P_{\max} , open-circuit voltage V_{OC} , internal resistance R_{in} and ΔT of n-Ag₂Se/p-Bi₂Te₃ module 2 during duration measurements at ΔT of ~ 85 K.

Fig. 4 Cooling performance. (a) measured maximum cooling temperature difference (ΔT_{\max}), (b) current dependent maximum cooling power ($Q_{c\max}$) and (c) coefficient of performance (COP), as well as (d) maximum COP as a function of different ΔT for the modules⁴⁷. (e) Cold-side temperature (T_c), hot-side temperature (T_h) and corresponding cooling temperature difference (ΔT) of n-Ag₂Se/p-Bi₂Te₃ module 2 at a given current I of 4.3 A during duration tests for 34 days, showing good stability. During the cooling duration tests, we run the test once per day for a continuous period of 22 hours.

Reviewer #3 (Remarks to the Author):

This work presents the fabrication and performance evaluation of thermoelectric modules based on n-type Ag_2Se . While recent thermoelectric research on Ag_2Se has primarily focused on its film performance and devices, I find this study's emphasis on Ag_2Se bulk modules to be particularly intriguing. This work systematically investigated the toughness of bulk Ag_2Se compared to Bi_2Te_3 , which is interesting and informative for future researches. The results demonstrate that Ag_2Se , with its superior bending and compressive strength, indeed represents a tougher alternative to n- Bi_2Te_3 . The contact resistivity here is low. I believe this may mark the first demonstration as I didn't find any literature reporting contact resistivity in bulk Ag_2Se device. The n- $\text{Ag}_2\text{Se}/\text{p-Bi}_2\text{Te}_3$ module seems to be the first demonstration as well, and the performance is good. With these important results, I recommend that this work be published in Nature Communications after addressing the technical concerns

1. Paragraph 3 in part "Results and discussion": "...The transport properties can be well described by a single parabolic band (SPB) model with an acoustic scattering (Figs. S3b and S3c).....". It is better to include the logarithmic relationship between temperature and mobility in Fig. S3 to illustrate acoustic scattering in Ag_2Se .

Response and revision: Thanks for the suggestion. We revised Fig. S3a to include a dotted line illustrating a nearly $T^{-1.5}$ dependence of hall mobility. This supports the reasonable use of SPB model with a dominant charge scattering by acoustic phonons.

2. We generally consider n-type Bi_2Te_3 to be more brittle compared to p-type Bi_2Te_3 . Why is it that here, the n-type Bi_2Te_3 exhibits higher bending and compressive strength?

Response and revision: Commercial Bi_2Te_3 -based materials are typically prepared by the zone-melting method, and zone-melted n-type Bi_2Te_3 indeed has inferior mechanical properties compared to the p-type. However, the commercial Bi_2Te_3 (Model No. TEG1-712-0.14, from Xiamen X-Meritan Technology Co., LTD) used in this work were prepared by hot extrusion technique. It has been proven that the mechanical properties of Bi_2Te_3 , including both microhardness and compressive strength, can be significantly improved by the hot extrusion technique [*Materials Today Physics*, 2023, 32, 101035]. And, after hot extrusion, the mechanical properties of n-type Bi_2Te_3 are better than those of p-type [see Fig.4 in *Materials Today Physics*, 2023, 32, 101035]. We revised the manuscript to include a brief clarification on this issue (line 44, page 2).

3. Geometry design is also important for maximizing the performance of thermoelectric module. Why did the authors select a leg size of $2 \times 2 \times 4 \text{ mm}^3$? Please provide further explanation.

Response and revision: Thanks for the suggestion. We carried out a numerical simulation to locate the optimal leg geometry for the $\text{Ag}_2\text{Se}/\text{p-Bi}_2\text{Te}_3$ module operating at a hot-side temperature (T_h) of 370 K and a cold-side temperature (T_c) of 285 K. The total cross-sectional area (A_{pn}) of a pair of legs is fixed to 8 mm^2 , with each leg having a cross-sectional area of A_p for p-type and A_n for n-type. H is the height of legs. As can be seen in Fig. S14, when $1 \leq A_p/A_n \leq 4$ and $H/A_{pn} > 0.25$, over 90% of the theoretical optimal efficiency can be achieved. In this work, for a more convenient fabrication, we chose the same size of $2 \times 2 \times 4 \text{ mm}^3$ for n- Ag_2Se as the commercial Bi_2Te_3 legs, corresponding to $A_p/A_n = 1$ and $H/A_{pn} = 1$ which lies within the above optimal range yet the device performance still has room for further improvement through geometric optimization. We added Fig. S14 in supplementary information with a brief explanation on this issue (line 24, page 3 in main text).

Fig. S14 Numerically simulated maximum efficiency (η_{\max}) as a function of the p- to n-type ratio in legs' cross-sectional area (A_p/A_n) and the ratio of height to the total cross-sectional area of a pair of legs (H/A_{pn}) for the n-Ag₂Se/p-Bi₂Te₃ module. Over 90% of the theoretical optimal efficiency can be achieved when $1 \leq A_p/A_n \leq 4$ and $H/A_{pn} > 0.25$.

4. Fig. S8 is not mentioned in the manuscript or the supplementary.

Response and revision: We have now mentioned Fig.S9 (Fig. S8 in the original version of supplementary) in the revised manuscript (line75, page 2).

5. The internal resistances of the Ag₂Se-based module and the commercial Bi₂Te₃ module differ. However, the optimized currents for these two modules appear to be the same, as shown in Figure 4a and Figure S8. Any explanation on this issue?

Response and revision: Thanks for the comments. The optimized current for cooling ΔT_{\max} is determined by $I_{\text{opt}} = S_{np} T_c / R_{\text{in}}$, where I_{opt} is the optimized current when temperature difference reaches maximum value, S_{np} is the absolute value of total Seebeck coefficient from n- and p- legs, T_c is the cold-side temperature and R_{in} is the internal resistance of the module. Thus, both S_{np} and R_{in} influence I_{opt} . Although the R_{in} of Ag₂Se-based module is obviously lower than that of commercial Bi₂Te₃ module (Fig. S9), n-Ag₂Se also has lower Seebeck coefficient compared to n-Bi₂Te₃. Therefore, the I_{opt} for these two modules appear to be close.

REVIEWERS' COMMENTS

Reviewer #1 (Remarks to the Author):

The revised manuscript can be published in Nature Communication now.

Reviewer #3 (Remarks to the Author):

It is acceptable in current revision

Responses to the reviewers' comments

Reviewer #1 (Remarks to the Author):

The revised manuscript can be published in Nature Communication now.

Response: We thank the reviewer for the time and the recommendation.

Reviewer #3 (Remarks to the Author):

It is acceptable in current revision.

Response: We thank the reviewer for the time and the recommendation.